## 1 Time shift between precipitation and evaporation has more

## 2 impact on annual streamflow variability than the elasticity of

## potential evaporation

4

3

- Vazken Andréassian\*,1, Guilherme Mendoza Guimarães1, Alban de Lavenne1, Julien
- Lerat<sup>2</sup>

- <sup>1</sup> Université Paris-Saclay, INRAE, HYCAR Research Unit, Antony, France
- <sup>2</sup> CSIRO, Canberra, Australia
- \*Corresponding author: Vazken Andréassian, vazken.andreassian@inrae.fr

#### **Abstract**

One of the most basic questions asked of hydrologists is the quantification of catchment response to climatic variations, i.e., the variations around the average annual flow given the climatic anomaly of a particular year. This paper presents an analysis based on 4122 catchments from four continents, where we investigate how annual streamflow variability depends on climate variables – rainfall and potential evaporation – and on the synchronicity between precipitation and potential evaporation. We use catchment data to verify the existence of this link and show that, in all countries and under the main climates represented, anomalies in this synchronicity are the second most important factor to explain annual streamflow anomalies, after precipitation, but before potential evaporation. Introducing the synchronicity between precipitation and potential evaporation as an independent variable improves the prediction of annual streamflow variability with an average additional explained variance of 6 % globally.

2627

28

**Keywords:** annual streamflow anomalies, elasticity, sensitivity, seasonality

#### **Notations**

- We deal in this paper with three hydrological fluxes: precipitation  $(P_n)$ , streamflow  $(Q_n)$
- and potential evaporation  $(E_{0n})$ . The three fluxes are computed at catchment scale,

expressed in millimeters per year, and represent annual totals (index n refers to the year in question). We use a hydrological year from October 1<sup>st</sup> of year n-1 to September 30<sup>th</sup> of year n in the Northern hemisphere and from April 1<sup>st</sup> of year n to March 31<sup>st</sup> of year n+1 in the Southern hemisphere. Anomalies (of P, Q and  $E_0$ ), noted  $\Delta$ , are computed as the difference between the annual value and the long-term average value, i.e.,  $\Delta Q_n = Q_n - \bar{Q}$ ,  $\Delta P_n = P_n - \bar{P}$ , etc.

## 1 Introduction

#### 1.1 On the climate elasticity of streamflow

To assess the impact of climate change on water resources, hydrologists aim to quantify the amount of change in catchment flow when climatic conditions vary. The ratio between changes in streamflow and climate is formally defined as the climate elasticity of streamflow (Schaake and Liu, 1989). The hydrological literature and common sense both suggest that the best factor explaining the changes in annual streamflow is the annual precipitation anomaly (e.g., Pardé, 1933a; Leopold, 1974). In addition, many elasticity studies have also considered the anomaly of potential evaporation, although it is usually only weakly statistically significant in regression studies. In this paper, we focus on a third explanatory variable that quantifies the synchronicity between precipitation and potential evaporation within the year.

#### 1.2 Linear models to predict streamflow anomalies

There is an abundance of literature concerning elasticity studies in hydrology, and our work builds upon the earlier empirical (i.e., measurement-based) studies of Sankarasubramanian et al. (2001), Chiew (2006), and Andréassian et al. (2016). Here, we follow the same principle and use linear regression models based on measured annual data to evaluate the climate elasticity of streamflow. An alternative approach to estimating climate elasticities would involve using hydrological models of varying complexities (e.g., Koster and Suarez, 1999). However, even if models are powerful investigative tools, they also rely on restrictive assumptions that often limit their credibility outside their calibration range. This can be particularly problematic in a large-scale study on the impact of climate change. Thus, we favored an approach introducing

the minimal number of hydrological assumptions, hence a linear regression that also has the advantage of being mathematically extremely simple.

## 1.3 The synchronicity between precipitation and potential evaporation impacts annual streamflow totals

The fact that the time shift between precipitation and potential evaporation, hereon referred to as "climatic synchronicity", has a hydrological impact has been known for a long time, as shown by a few precursors on this topic. For example, in 1933b Pardé published a classic paper dedicated to the average flow of rivers, where he underlined that "for identical values of precipitation and temperature, everything else being equal, the runoff coefficient Q/P will be smaller where the larger part of precipitation falls during the warm season". Similarly, Coutagne and de Martonne (1935) discussed formulas for annual streamflow, and underlined that formulas based only on the humidity ratio P/E<sub>0</sub> are deficient, because they fail to account for "the distribution of precipitations between seasons, in particular, in the temperate zone, between the warm and the cold season. Of two years of equal precipitation, the year which will receive the most part in summer will produce the less annual flow". Additional classical studies include Thornthwaite (1948), who proposed to classify climates initially with two indices (one characterizing the periods of water surplus and the other the periods of water deficiency), which he subsequently combined into a single index. Also, Turc introduced in 1954 his famous formula for long-term actual evaporation. At the very end of his paper, he wrote that "the most urgent improvement" to his actual evaporation formula should be the introduction of the "distribution of precipitations and of the temperature changes within the year." Recent studies have also discussed the impact of climate seasonality on water balance, based on either theoretical or empirical approaches. Among the theoretical studies, Dooge (1992) presented catchment yield curves where he introduced as a parameter the length of the dry season. Milly (1994) proposed a theoretical computation of actual evaporation based on the seasonality of the aridity index. Yokoo et al. (2008) made theoretical computations on the difference between in-phase and out-of-phase regimes of precipitation and potential evapotranspiration. Additionally, Roderick and Farguhar (2011), Feng et al. (2012), Donohue et al. (2012), Berghuijs et al. (2014) and Jawitz et al. (2022) all made notable developments. Among the empirical studies, Potter et al. (2005) quantified the impact of rainfall seasonality on mean annual

water balance in Australia. Hickel and Zhang (2006) discussed the antagonistic effects of climate seasonality and soil moisture storage. More recently, de Lavenne and Andréassian (2018) proposed a synchronicity index to characterize the phase difference between precipitation and potential evaporation, and Feng et al. (2019) proposed an index of asynchronicity for Mediterranean climates.

#### 1.4 Purpose of the paper

In this paper, we aim to improve the prediction of streamflow elasticity by introducing anomalies in synchronicity between precipitation and potential evaporation as a predictor, alongside variability in rainfall and potential evapotranspiration. Our study is based solely on data analysis, and uses only linear regression models.

## 2 Test catchments

#### 2.1 Origin of the dataset

As presented in Table 1, we use catchments from nine countries to base our analysis on a wide range of climates.

Table 1. Origin of the catchments used in this paper

| Country        | Number of catchments selected | Number of catchments available in the original dataset | Dataset                   | Reference                |
|----------------|-------------------------------|--------------------------------------------------------|---------------------------|--------------------------|
| Australia      | 546                           | 561                                                    | Camels-AUS                | Fowler et al. (2024)     |
| Brazil         | 636                           | 734                                                    | Cabra                     | Almagro et al. (2021)    |
| Denmark        | 202                           | 304                                                    | Camels-DK                 | Liu et al. (2024)        |
| France         | 628                           | 654                                                    | Camels-FR                 | Delaigue et al. (2024)   |
| Germany        | 1094                          | 1555                                                   | Camels-DE                 | Loritz et al. (2024)     |
| Sweden         | 152                           | 158                                                    | Selection by G. Lindström | de Lavenne et al. (2022) |
| Switzerland    | 73                            | 331                                                    | Camels-CH                 | Höge et al. (2023)       |
| United Kingdom | 136                           | 670                                                    | Camels-UK                 | Coxon et al. (2020)      |
| USA            | 655                           | 672                                                    | Camels-US                 | Addor et al. (2017)      |

The total number of catchments is 4122, for a total of 162,005 station-years (the average length of catchment time series is 39 years). We use hydrological years as defined in the Notations section.

#### 2.2 Catchment selection

128

140

The catchments used in this paper were selected from several datasets indicated in 115 Table 1 and represent approximately 75% of the original catchments. Our catchment 116 selection was based on three criteria: record length, catchment memory and regulation 117 degree. First, we only selected catchments that had more than 20 complete 118 hydrological years. Second, we selected catchments that exhibit minimal interannual 119 memory ("memory" as defined by de Lavenne et al., 2022). This criterion was needed 120 because the equation used here to estimate streamflow elasticity is only hydrologically 121 warranted for catchments displaying minimum interannual memory, thus allowing a 122 straightforward computation of annual elasticity coefficients, based only on annual 123 average values. Finally, catchments identified as significantly regulated by reservoirs 124 were removed. This identification was done by either asking the datasets authors, or, 125 where the information was available, by setting a limit equal to 10 mm equivalent 126 volume storage in dams). For Switzerland, the list of almost natural catchments 127 published by Muelchi et al. (2022) was utilized.

#### 2.3 Climatic inputs

- Where several precipitation products were available in the original dataset, we used
- the product recommended by dataset authors as being of the best quality, while
- avoiding precipitation data based exclusively on satellite estimates.
- In the original datasets, potential evaporation was computed with a variety of different
- formulas (Makkink, Morton, FAO-56, Penman-Monteith, Hargreaves, Oudin, etc.). For
- the sake of homogeneity, we recomputed it (at the daily time step) for all catchments
- using the formula proposed by Oudin et al. (2005), which requires only extraterrestrial
- radiation and air temperature. This formula was selected for two reasons: first, it could
- be computed, given the available data, for all datasets, and it has been widely used
- worldwide and appears appropriate (while of course not perfect) for describing
- atmospheric evaporative demand.

#### 2.4 Characteristics of the catchment set

- In our dataset, the aridity index, computed as  $E_0/P$ , ranges from 0.1 to 6.3, with a first
- quartile of 0.6 and a third quartile of 1.0. The mean and the median of the aridity index
- are both 0.8. In order to assess the generality of the results, we will discuss them at

the country scale and also by climatic classes following the Köppen-Geiger classification (see e.g., Peel et al. 2007 and Table 2). Note that we only give numerical results for the climatic zones with more than 100 catchments.

Table 2. Main climatic zones (in the sense of the Köppen-Geiger classification) represented in our dataset (we present the zones counting more than 100 catchments)

| Köppen-Geiger<br>zone | Name                                                    | Number of catchments |  |  |
|-----------------------|---------------------------------------------------------|----------------------|--|--|
| Aw                    | Tropical savanna climate with dry winter                | 344                  |  |  |
| Cfa                   | Temperate climate without dry season with hot summer    | 364                  |  |  |
| Cfb                   | Temperate climate without dry season with warm summer   | 1746                 |  |  |
| Csa                   | Temperate climate with dry and hot summers              | 196                  |  |  |
| Dfb                   | Continental climate without dry season with warm summer | 956                  |  |  |
| Dfc                   | Continental climate without dry season with cold summer | 132                  |  |  |

Finally, Figure 1 presents the 4122 catchments of our dataset with two variants of the Turc-Budyko non-dimensional graph. On the left-hand graph, each catchment corresponds to one point, with coordinates representing the average aridity on the x-axis and 'actual evaporation' yield, computed as (P-Q)/P, on the y-axis. On the right-hand graph, each catchment is represented by a single point, with coordinates indicating the average humidity on the x-axis and the average streamflow yield, computed as Q/P, on the y-axis.

Figure 1: Representation of the 4122 catchments in two equivalent forms of the Turc-Budyko non-dimensional space. The solid blue line corresponds to the water limit (Q=P), and the orange line corresponds to the energy limit (Q=P- $E_0$ ). On the left, an additional limit (dotted blue line) is sometimes improperly referred to as "water limit" in the literature, but it only corresponds to the physical limit (Q=0), when one estimates the actual evaporation as the difference between discharge and precipitation. The catchments that are beyond the orange line (i.e., above on the left and below on the right) are "leaky" (in the sense that they contribute to the recharge of a regional aquifer) and those which are beyond the blue line (i.e., below on the left and above on the right) are "gaining" in the sense of a karstic catchment which would drain a larger than specified catchment (note that in a few cases, data uncertainties might also cause catchments to be beyond the limits).

#### 3 Method

#### 3.1 Computation of the synchronicity of precipitation and potential evaporation

In this paper, we utilize a modified version of the seasonality index introduced by de Lavenne and Andréassian (2018): a detailed discussion of the reasons for this change is provided in the Appendix. The objective of this index ( $\Lambda$ ) is to characterize the synchronicity between precipitation P and potential evaporation  $E_0$  at the annual time step. For each year n, we define the part of annual precipitation that is the most easily accessible to evaporation (i.e., neutralizable by evaporation) as in Eq. 1 and Figure 2:

$$synchronous\ P-E_0\ amount=\sum_{m=1}^{12}min\left(P_{m,n}\ ,E_{0\,m,n}\right)$$
 Eq. 1

where the index *m* refers to the calendar month

Figure 2. two series of precipitation and potential evaporation at catchment scale: the part of precipitation that is the most easily accessible to evaporation is illustrated in hatched pattern

The percentage of easily neutralizable precipitation is then defined as Eq. 2, and the percentage of easily neutralizable potential evaporation as Eq. 3.

$$\lambda_{1,n} = \frac{\sum_{m=1}^{12} min\left(P_{m,n} \text{ , } E_{0_{m,n}}\right)}{P_n}$$
 Eq. 2

$$\lambda_{2,n} = \frac{\sum_{m=1}^{12} min\left(P_{m,n} \text{ , } E_{0_{m,n}}\right)}{E_{0_n}}$$
 Eq. 3

Because both ratios belong to the interval [0,1], their geometric average will also be within the same range (Eq. 4).

$$\lambda_{3,n} = \sqrt{\lambda_{1,n}\lambda_{2,n}} = \frac{\sum_{m=1}^{12} min\left(P_{m,n} \text{ , } E_{0_{m,n}}\right)}{\sqrt{P_n \, E_{0_n}}}$$
 Eq. 4

Finally, the index  $\Lambda$  rescales and combines  $\lambda_1$  and  $\lambda_2$  into a single quantity, expressed in mm/yr, representing the average ratio of neutralizable precipitation and neutralizable

potential evaporation as shown in Eq. 5. For two years with the same annual amounts of precipitation and potential evaporation,  $\Lambda$  will reach higher values when P and  $E_0$  are synchronous, and lower values when they are out of phase.

$$\varLambda_{n} = \lambda_{3,n} * \bar{P} = \frac{\sum_{m=1}^{12} min\left(P_{m,n}, E_{0_{m,n}}\right)}{\sqrt{P_{n} * E_{0_{n}}}} * \bar{P}$$
 Eq. 5

#### 196 3.2 Computation of streamflow elasticities

To compute the streamflow elasticities, we solve two linear equations given by Eq. 6 and Eq. 7.

$$\Delta Q_n = e_{Q/P} \Delta P_n + e_{Q/E_0} \Delta E_{0_n} \tag{Eq. 6}$$

$$\Delta Q_n = e_{Q/P} \Delta P_n + e_{Q/E_0} \Delta E_{0n} + e_{Q/A} \Delta \Lambda_n \tag{Eq. 7}$$

- Where  $\Delta Q_n$  (respectively  $\Delta P_n$ ,  $\Delta E_{0n}$ ,  $\Delta \Lambda_n$ ) represents the deviation from the mean annual value (anomaly) for variable Q (respectively P,  $E_0$ ,  $\Lambda$ ) in mm/y and  $e_{Q/P}$ ,  $e_{Q/E_0}$
- and  $e_{Q/\Lambda}$  represent the elasticity of streamflow with respect to P,  $E_0$ , and
- $\Lambda$  (dimensionless).

- Eq. 6 represents the classical approach to elasticity computation (Andréassian et al.,
- 2016), while Eq. 7 represents the original contribution of this paper, and aims at
- determining how far climatic synchronicity explains annual streamflow variability.
- The elasticities in Eq. 6 and Eq. 7 are estimated via ordinary least squares (OLS). More
- complex statistical models such as generalized least squares are not required because
- the selected catchments do not exhibit interannual memory, as explained in the data
- section. This absence of interannual memory guarantees the lack of autocorrelation in
- annual streamflow, which is an important statistical assumption for OLS. Additionally,
- we chose a p-value threshold of 0.05 for all the discussion of results. We compute
- elasticity coefficients between anomalies of equal dimensions (in mm/y), and not
- between relative anomalies (in %) because with the anomalies expressed in mm/y the
- physically-plausible range is known: [0,1] for  $e_{O/P}$ , [-1,0] for  $e_{O/E_0}$  and  $e_{O/A}$ . Finally, Eq.
- 6 and Eq. 7 were solved on a catchment-by-catchment basis, i.e., we computed 4122
- distinct regressions.
- Figure 3 illustrates this catchment-based computation using the example of the
- Meurthe River at Raon-l'Étape (727 km²). For this catchment, annual streamflow

anomalies exhibit a well-defined dependency on both precipitation and synchronicity anomalies, with the dependency on potential evaporation anomaly being very weak.

Figure 3. Example of an elasticity plot for the Meurthe River at Raon-l'Étape (A615103001): each point corresponds to one hydrological year (for this catchment, 36 hydrological years were available, from 1975 to 2021). The Pearson correlations of  $\Delta Q$  with  $\Delta P$ ,  $\Delta E_0$  and  $\Delta \Lambda$  are respectively 0.87, -0.02 and -0.78

The visual impression of Figure 3 is confirmed by the results of the linear regressions of Eq. 6 and Eq. 7 in Table 3. Values of the Student's t-test indicate that precipitation has a dominant contribution, while the contribution of potential evaporation is not statistically significant. The introduction of synchronicity increases the R<sup>2</sup> from 0.75 to 0.80.

Table 3. Climate elasticity coefficients computed with and without the inclusion of the synchronicity variable  $\Lambda$  for the example catchment (La Meurthe at Raon-l'Étape)

| Formulation                                          | $e_{Q/P}$ [-] | p-value for $e_{\mathit{Q/P}}$ | $e_{Q/E_0}$ [-] | $p	ext{-}value \ for e_{Q/E_0}$ | $e_{Q/\varLambda}$ [-] | p-value for $e_{Q/arLet}$ | R²   |
|------------------------------------------------------|---------------|--------------------------------|-----------------|---------------------------------|------------------------|---------------------------|------|
| $\Delta Q = f(\Delta P, \Delta E_0)$                 | 0.52          | 

Figure 4 Scatter plots, for each country, between streamflow anomalies  $\Delta Q$ , and: precipitation anomalies  $\Delta P$  (left), potential evaporation anomalies  $\Delta E_0$  (middle) and synchronicity index anomalies  $\Delta \Lambda$  (right). Each point represents one station-year. Above each scatter plot, we provide the corresponding Pearson correlation

Figure 4. (continuation)

Figure 5. Scatter plots, for the main climate classes, between streamflow anomalies  $\Delta Q$ , and: precipitation anomalies  $\Delta P$  (left), potential evaporation anomalies  $\Delta E_0$  (middle) and synchronicity index anomalies  $\Delta A$  (right). Each point represents one station-year. Above each scatter plot, we provide the corresponding Pearson correlation

#### 4.2 Overall results by catchment

We now analyze the results obtained for each of the 4122 catchments. Table 4 shows the statistics of the individual regressions for the classical case when synchronicity is not included as a predictor. This analysis reveals that for all countries and all climate groups, the precipitation elasticity of streamflow is almost always significant at the 0.05 level. On the other hand, the potential evaporation elasticity of streamflow is not frequently significant at the 0.05 level. In addition, the regression identifies physically realistic precipitation elasticity values (between 0 and 1) for almost all catchments (93% worldwide, and a minimum of 80% across different groupings), whereas potential evapotranspiration elasticity is frequently physically unrealistic with only 6% of values in the range [0, 1] globally.

Table 4. Linear regression results by country and climate type for Eq. 6 when regression uses two independent variables P and  $E_0$  to explain streamflow anomaly

| Region or climate | Total number of | catchmer                      | tage of<br>nts where<br>was           | Percen catchmer $e_{Q/E_0}$   | Mean<br>adjusted                    |      |  |
|-------------------|-----------------|-------------------------------|---------------------------------------|-------------------------------|-------------------------------------|------|--|
| class             | catchments      | significant at the 0.05 level | significant and in<br>the range [0,1] | significant at the 0.05 level | significant and in the range [-1,0] | R²   |  |
| By country        |                 |                               | <u> </u>                              |                               |                                     |      |  |
| Australia         | 546             | 100%                          | 97%                                   | 18%                           | 9%                                  | 0.67 |  |
| Brazil            | 636             | 95%                           | 86%                                   | 12%                           | 4%                                  | 0.61 |  |
| Denmark           | 202             | 100%                          | 100%                                  | 9%                            | 0%                                  | 0.51 |  |
| France            | 628             | 100%                          | 93%                                   | 21%                           | 7%                                  | 0.71 |  |
| Germany           | 1094            | 94%                           | 93%                                   | 18%                           | 9%                                  | 0.47 |  |
| Sweden            | 152             | 100%                          | 87%                                   | 20%                           | 7%                                  | 0.65 |  |
| Switzerland       | 73              | 100%                          | 86%                                   | 8%                            | 0%                                  | 0.75 |  |
| UK                | 136             | 99%                           | 89%                                   | 25%                           | 2%                                  | 0.75 |  |
| USA               | 655             | 99%                           | 95%                                   | 9%                            | 4%                                  | 0.65 |  |
| By climate        | class           |                               |                                       |                               |                                     |      |  |
| Aw                | 344             | 93%                           | 91%                                   | 16%                           | 7%                                  | 0.60 |  |
| Cfa               | 364             | 100%                          | 90%                                   | 3%                            | 0%                                  | 0.66 |  |
| Cfb               | 1746            | 98%                           | 94%                                   | 18%                           | 7%                                  | 0.60 |  |
| Csa               | 196             | 99%                           | 96%                                   | 7%                            | 1%                                  | 0.67 |  |
| Dfb               | 956             | 96%                           | 94%                                   | 21%                           | 9%                                  | 0.56 |  |
| Dfc               | 132             | 99%                           | 80%                                   | 29%                           | 10%                                 | 0.71 |  |
| World             | 4122            | 97%                           | 93%                                   | 16%                           | 6%                                  | 0.61 |  |

Aw - Tropical savanna climate with dry winter, Cfa – Temperate climate without dry season with hot summer, Cfb – Temperate climate without dry season with warm summer, Csa – Temperate climate with dry and hot summers, Dfb – Continental climate without dry season with warm summer, Dfc – Continental climate without dry season with cold summer

Table 5 presents the same statistics, when the synchronicity anomaly  $(\Delta \Lambda_n)$  is introduced into the elasticity regression (Eq. 7). This analysis shows that the average

efficiency of the regression equation increases across all countries and climate groups (see also Figure 6). While an increase is expected when an additional predictor is added to a regression, please note that we are presenting adjusted R<sup>2</sup> values, which are designed to take that issue into account. The average additional explained variance lies in the range 3 %-10 % (6 % globally), depending on the group, and we consider it a noticeable improvement. Additionally, the synchronicity anomaly  $(\Delta \Lambda_n)$ provides a significant contribution to the regression for 64 % of the catchments, compared to only 23 % for potential evaporation). More important, the introduction of the synchronicity anomaly  $(\Delta \Lambda_n)$  does not modify the significance of the other two elasticity coefficients  $e_{O/P}$  and  $e_{O/E_0}$ . A slight increase is observed in the proportion of catchments where  $e_{Q/E_0}$  coefficient is significant at the 0.05 level (from 16 % to 23 %). Moreover, the utilization of  $\Delta \Lambda_n$  does not degrade the physical realism of the elasticity coefficients  $e_{O/P}$  and  $e_{O/E_0}$ . Once again, a slight increase is observed in the proportion of catchments where  $e_{O/P}$  coefficient is significant and in the physical range [0,1] (from 93 % to 94 %), and where  $e_{O/E_0}$ coefficient is significant at the 0.05 level and in the physical range [-1,0] (from 6 % to 11 %). Finally, only two countries (Switzerland and Brazil) and one climate type (*Dfc* – Continental climate without dry season with cold summer) showed lower relevance of the synchronicity index compared to other regions. We attribute this reduced relevance in Switzerland and climate zone Dfc to the essentially energy-limited nature of the catchments as our selection criteria for Switzerland prioritized high-elevation catchments with minimal anthropogenic impact (see also the Discussion section and Figure 9). Last, note that in all groupings except Dfc, the number of catchments where  $e_{O/A}$  is significant at the 0.05 level exceeds that where the  $e_{O/E_0}$  coefficient is significant

at the same level.

| Country         | Total<br>number of<br>catchment<br>s | Percentage of catchments where $e_{O/P}$ was |                                              | Percentage of catchments where $e_{Q/E_0}$ was |                                               | Percentage of catchments where $e_{Q/\Lambda}$ was |                                               | Mea<br>n<br>adj. | Mean<br>adj.<br>R²<br>from |
|-----------------|--------------------------------------|----------------------------------------------|----------------------------------------------|------------------------------------------------|-----------------------------------------------|----------------------------------------------------|-----------------------------------------------|------------------|----------------------------|
|                 |                                      | significan<br>t at the<br>0.05 level         | significan<br>t and in<br>the range<br>[0,1] | significan<br>t at the<br>0.05 level           | significan<br>t and in<br>the range<br>[-1,0] | significan<br>t at the<br>0.05 level               | significan<br>t and in<br>the range<br>[-1,0] | R²               | Tabl<br>e 4                |
| By country      |                                      |                                              |                                              |                                                |                                               |                                                    |                                               |                  |                            |
| Australia       | 546                                  | 100%                                         | 98%                                          | 38%                                            | 20%                                           | 87%                                                | 83%                                           | 0.76             | 0.67                       |
| Brazil          | 636                                  | 90%                                          | 84%                                          | 13%                                            | 5%                                            | 25%                                                | 22%                                           | 0.64             | 0.61                       |
| Denmark         | 202                                  | 100%                                         | 100%                                         | 6%                                             | 0%                                            | 44%                                                | 44%                                           | 0.56             | 0.51                       |
| France          | 628                                  | 100%                                         | 96%                                          | 30%                                            | 13%                                           | 82%                                                | 79%                                           | 0.77             | 0.71                       |
| Germany         | 1094                                 | 97%                                          | 97%                                          | 27%                                            | 16%                                           | 79%                                                | 76%                                           | 0.57             | 0.47                       |
| Sweden          | 152                                  | 100%                                         | 90%                                          | 24%                                            | 5%                                            | 41%                                                | 38%                                           | 0.69             | 0.65                       |
| Switzerlan<br>d | 73                                   | 96%                                          | 82%                                          | 8%                                             | 0%                                            | 22%                                                | 21%                                           | 0.76             | 0.75                       |
| UK              | 136                                  | 99%                                          | 90%                                          | 41%                                            | 11%                                           | 62%                                                | 59%                                           | 0.81             | 0.75                       |
| USA             | 655                                  | 99%                                          | 96%                                          | 11%                                            | 5%                                            | 57%                                                | 52%                                           | 0.71             | 0.65                       |
| By climate      | class                                |                                              |                                              |                                                |                                               | •                                                  |                                               |                  |                            |
| Aw              | 344                                  | 90%                                          | 88%                                          | 18%                                            | 10%                                           | 42%                                                | 40%                                           | 0.67             | 0.60                       |
| Cfa             | 364                                  | 98%                                          | 91%                                          | 9%                                             | 1%                                            | 51%                                                | 47%                                           | 0.74             | 0.66                       |
| Cfb             | 1746                                 | 99%                                          | 96%                                          | 28%                                            | 14%                                           | 76%                                                | 74%                                           | 0.71             | 0.60                       |
| Csa             | 197                                  | 99%                                          | 97%                                          | 17%                                            | 2%                                            | 43%                                                | 37%                                           | 0.73             | 0.67                       |
| Dfb             | 956                                  | 98%                                          | 96%                                          | 27%                                            | 14%                                           | 68%                                                | 65%                                           | 0.66             | 0.56                       |
| Dfc             | 132                                  | 98%                                          | 82%                                          | 30%                                            | 8%                                            | 30%                                                | 29%                                           | 0.76             | 0.71                       |
| World           | 4122                                 | 97%                                          | 94%                                          | 23%                                            | 11%                                           | 64%                                                | 61%                                           | 0.67             | 0.61                       |

Aw - Tropical savanna climate with dry winter, Cfa – Temperate climate without dry season with hot summer, Cfb – Temperate climate without dry season with warm summer, Csa – Temperate climate with dry and hot summers, Dfb – Continental climate without dry season with warm summer, Dfc – Continental climate without dry season with cold summer

#### 5 Discussion

Figure 6 illustrates the improvement in explanatory capacity of the regressions due to the introduction of the synchronicity anomalies. While considerable variability exists, and some catchments show equivalent performance between the two regression models (indicated by points on the 1:1 line), the graph confirms that for many catchments (approximately 66 % of the dataset, where  $e_{Q/\Lambda}$  was significant at the 0.05 level), accounting for synchronicity anomalies visibly improves the efficiency of the linear regression. Because the adjusted R² shows the same trend as the classical R², this is clearly not a simple effect of the increase of independent variables in the regression.

Figure 6. Comparison of the performances of the 2-parameter streamflow elasticity model (Eq. 6, which does not account for P- $E_0$  synchronicity) and the 3-parameter model (Eq. 7, which does). Each point represents one of the 4122 catchments of our dataset. The solid line represents the median, and the dashed lines represent the first and the third quartiles. As measure of efficiency, we use the  $R^2$  on the left plot and the adjusted  $R^2$  on the right one

In Figure 7, we summarize the significance of the  $e_{Q/\Lambda}$  coefficient across all the Köppen climate classes represented in our dataset:  $e_{Q/\Lambda}$  is significant at the 0.05 level for 50 % of the catchments in 11 classes (representing 79% of the catchments).

Figure 7. significance of the P-E<sub>0</sub> synchronicity anomalies by Köppen climate class: the dashed area represents the proportion of catchments for which synchronicity was not deemed significant

Figure 8 shows the geographic distribution of the catchments where the P-E<sub>0</sub> synchronicity had a significant contribution to explain streamflow anomalies (with a p-value threshold of 0.05). The map brings further elements to Table 5 and illustrates that there are sub-regions where the coefficient  $e_{Q/\Lambda}$  is mostly not significant at the 0.05 level. Based on our knowledge of the climatic specificities of each country, this seems to be possibly correlated to higher rainfall (cf. the Danish dataset, with the particular behavior of the West of Jutland, the case of Florida in the US, the case of the Scottish catchments in Great Britain) and/or to colder areas (cf. the Swiss, Swedish and US datasets).

Figure 8. map of the 4122 catchments used in this study, each catchment is represented by either a circle (where the P-E<sub>0</sub> synchronicity anomalies had a significant contribution to explain streamflow anomalies) or a cross (where it was not significant at the 0.05 level). The color of circles and crosses corresponds to the Köppen climate classes

To verify this hypothesis, Figure 9 presents the p-values of the 4122  $e_{Q/\Lambda}$  coefficients as a function of the humidity index P/E<sub>0</sub>. This graph clearly indicates that most of the humid catchments (Humidity index > 2) lack sensitivity to the P-E<sub>0</sub> seasonality, and this pattern is likely the main explanation for the geographical patterns observed in Figure 8.

Figure 9: distribution of the p-values of the 4122  $e_{Q/\Lambda}$  coefficients as a function of the humidity index P/E<sub>0</sub>. The red points represent the median, the bar represent the interquartile range, and the dashed line represents the 0.05 threshold.

#### 6 Conclusion

#### 6.1 Synthesis

In this paper, we investigated the dependency between streamflow elasticity and the synchronicity of precipitation and potential evaporation, using a dataset of 4122 catchments located in Europe, Australia, North America and South America. Our analysis provided three main findings. First, we empirically verified the strong correlation among streamflow anomalies, annual precipitation anomalies, and synchronous P-E0 anomalies. Second, we demonstrated that the role of the synchronicity between P and E0 in explaining streamflow anomalies is significantly more important than that of E0 anomalies. Finally, we showed that introducing synchronicity between precipitation and potential evaporation as an additional predictor in the linear regression clearly improves the prediction of annual streamflow variability.

#### 6.2 Perspectives

Notwithstanding these positive results, some estimated elasticity values remain outside of their physically acceptable domain (i.e., [0,1] for  $e_{Q/P}$  and [-1,0] for  $e_{Q/E_0}$  and  $e_{Q/A}$ ). For precipitation elasticity ( $e_{Q/P}$ ), 93% of the catchments were within the physical range, out of a total of 97% where precipitation elasticity was significant. For potential

evaporation elasticity  $(e_{Q/E_0})$ , a lack of physical realism occurs in most of the cases (i.e., only 11% of the catchments were within the physical range, out of a total of 23% where potential evaporation elasticity was significant). This is very likely due to a sensitivity problem in the regression, which contributes to the difficulty in obtaining realistic elasticity coefficients. Finally, for synchronicity elasticity  $(e_{Q/A})$ , 61% of the catchments were within the physical range out of a total of 64% where synchronicity elasticity was significant. In the future, we aim to investigate alternative statistical models that could better constrain the elasticity coefficients within their physically realistic domain.

### 7 Acknowledgements

- The authors would like to acknowledge the many individuals that worked to make
- available the hydrological datasets used in this paper. Special thanks are due to the
- two anonymous reviewers, to the Editor, as well as to Charles Perrin and Guillaume
- Thirel for their reviews and suggestions and to Laurent Strohmenger for his help with
- the Köppen-Geiger classification.

## 416 **8 Funding**

- This research has been funded in part by the Agence Nationale de la Recherche
- (projects CIPRHES ANR-20-CE04-0009 and DRHYM ANR-22-CE56-0007).

#### 419 **9 Author contributions**

- VA: conceptualization and writing, GMG: computations, figures, discussion, writing
- (review and editing), AL: computations, discussion, figures, JL: discussion, writing
- (review and editing)

## 10 Competing interests

The authors declare that they have no conflict of interest.

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

| 522 | Peel, M.C., Finlayson, B.L., McMahon, T.A. 2007. Updated world map of the Köppen-                                                       |
|-----|-----------------------------------------------------------------------------------------------------------------------------------------|
| 523 | Geiger climate classification, Hydrol. Earth System Sci., 11(5):1633-1644.                                                              |
| 524 | https://doi.org/10.5194/hess-11-1633-2007                                                                                               |
| 525 | Potter, N.J., Zhang, L., Milly, P.C.D., McMahon, T.A., Jakeman, A.J., 2005. Effects of                                                  |
| 526 | rainfall seasonality and soil moisture capacity on mean annual water balance for                                                        |
| 527 | Australian catchments. Water Resour. Res. 41 (6).                                                                                       |
| 528 | https://doi.org/10.1029/2004wr003697                                                                                                    |
| 529 | Roderick, M.L., Farquhar, G.D., 2011. A simple framework for relating variations in                                                     |
| 530 | runoff to variations in climatic conditions and catchment properties. Water                                                             |
| 531 | Resour. Res., 47. https://doi.org/10.1029/2010WR009826                                                                                  |
| 532 | Sankarasubramanian, A., Vogel, R.M., Limbrunner, J.F., 2001. Climate elasticity of                                                      |
| 533 | streamflow in the United States. Water Resour. Res., 37(6): 1771-1781.                                                                  |
| 534 | https://doi.org/10.1029/2000wr900330                                                                                                    |
| 535 | Schaake, J., Liu, C., 1989. Development and application of simple water balance                                                         |
| 536 | models to understand the relationship between climate and water resources, New                                                          |
| 537 | Directions for Surface Water Modeling. IAHS Red Book series n°181,                                                                      |
| 538 | Wallingford, pp. 343-352. <a href="https://iahs.info/uploads/dms/7849.343-352-181-">https://iahs.info/uploads/dms/7849.343-352-181-</a> |
| 539 | Schaake-Jr.pdf                                                                                                                          |
| 540 | Thornthwaite, C.W., 1948. An approach toward a rational classification of climate.                                                      |
| 541 | Geog. Rev. 38 (1), 55–94. https://doi.org/10.2307/210739                                                                                |
| 542 | Turc, L. 1954. The water balance of soils: relationship between precipitations,                                                         |
| 543 | evaporation and flow (In French: Le bilan d'eau des sols: relation entre les                                                            |
| 544 | précipitations, l'évaporation et l'écoulement), Annales Agronomiques, Série A,                                                          |
| 545 | 491-595.                                                                                                                                |
| 546 | Yokoo, Y., Sivapalan, M., Oki, T. 2008. Investigating the role of climate seasonality                                                   |
| 547 | and landscape characteristics on mean annual and monthly water balances. J.                                                             |
| 548 | Hydrol. 357 (255–269). <a href="https://doi.org/10.1016/j.jhydrol.2008.05.010">https://doi.org/10.1016/j.jhydrol.2008.05.010</a>        |
| 549 |                                                                                                                                         |

# 551 **12 Appendix:** further details to justify our choice for the synchronicity index

There is no unique solution for choosing a measure of synchronicity between Precipitation and Potential Evaporation. In a previous paper (de Lavenne & Andréassian, 2018) we presented a non-dimensional index ( $\lambda$ ), defined as follows (Eq. 8):

$$\lambda = \frac{\sum_{m=1}^{12} min\left(P_{m,n}, E_{0_{m,n}}\right)}{\sum_{m=1}^{12} max\left(P_{m,n}, E_{0_{m,n}}\right)}$$
 Eq. 8

A reviewer of this paper remarked that our interpretation of this index did not hold in extreme cases. Thus, we modified it in order to improve its interpretability. We also tried to replace it with simpler versions, and we would like to present these alternatives in order to save time and effort for those who would like to keep working on this topic. The first simplification which was tested (called here S1) consisted in using directly the synchronous  $P - E_0$  amount:

$$S1(n) = \sum_{m=1}^{12} min\left(P_{m,n}, E_{0_{m,n}}\right)$$
 Eq. 9

below:

S1 was an interesting solution because it yielded directly a value in mm/y, without the need for rescaling, and it clearly represented the precipitation volume that was the most easily accessible to evaporation. In the linear regression, it did give very high average adjusted R² (world average of 0.67, the same as for the solution retained). The reason why we did not consider this solution was that there was a correlation between  $\Delta S1$  and  $\Delta P$  for many catchments (average correlation of +0.58 over the 4122 catchments, reaching +0.74 over the Australian catchments), and introducing two correlated variables in a regression equation is clearly bad statistical practice. To avoid this high correlation, we tested a normalization using annual precipitation, which we redimensionalized using the average interannual precipitation as in Eq. 10

$$S2(n) = \frac{\sum_{m=1}^{12} min\left(P_{m,n}, E_{0_{m,n}}\right)}{P_n} * \bar{P}$$
 Eq. 10

- The problem we found with S2 was that it yielded a constant value (equal to  $\bar{P}$ ) for
- many arid catchments, where for most of the years  $\frac{\sum_{m=1}^{12} (P_{m,n} \cap E_{0_{m,n}})}{P_n} = 1$  because
- $P_{m,n} \ll E_{0_{m,n}}$ .
- We also tested a normalization using annual potential evaporation, which we
- redimensionalized using the average interannual potential evaporation as in Eq. 11
- below:

$$S3(n) = \frac{\sum_{m=1}^{12} min\left(P_{m,n}, E_{0_{m,n}}\right)}{E_{0_n}} * \overline{E_0}$$
 Eq. 11

- But S3 behaved similarly as S1 (clearly because the  $\frac{\overline{E_0}}{E_{0n}}$  ratio is always close to 1), and
- the issue of having highly correlated values of  $\Delta S3$  and  $\Delta P$  reappeared.

- This is why we finally opted for combining S2 and S3 using a geometric average (which
- correlation with the annual P is low: -0.10 on average), which was then
- redimensionalized using the average interannual precipitation. This yielded  $\Lambda_n$ , which
- has the desired dimension (mm/y), and was used throughout this paper.

587

$$\varLambda_n = \frac{\sum_{m=1}^{12} min\left(P_{m,n}, E_{0_{m,n}}\right)}{\sqrt{P_n E_{0_n}}} * \bar{P}$$
 Eq.4