# Peer review of "Time shift between precipitation and evaporation has more"

_EGUsphere, 2025_

## Author Comment (AC2)

Dear Reviewer,

Thank you for reviewing our paper and for your encouraging remarks. We will address them all in the revised version. In this answer, we will only address the central remark of your review, i.e. that pertaining to the choice of the synchronicity index. We first discuss the new solution that we propose, and then analyze your suggestion (which we found to be linearily corrected to one of the alternatives presented in the appendix of the paper).

**1. Renaming our new synchronicity index or proposing a new one**

First of all, we agree with your computations, which clearly identify the actual behaviour of our present synchronicity index. The definition we gave ("$\lambda_n$ *represents the percentage of annual precipitation that is the most easily accessible to evaporation*") is approximative, and the examples you gave show it. It comes from the difficulty to describe with words the two members of the ratio: while it is easy to write that the numerator $\sum_{m=1}^{12} \left( P_{m,n} \cap E_{0\,m,n} \right)$ represents the quantity of precipitation that is the most accessible to evaporation, the denominator $\sum_{m=1}^{12} \left( P_{m,n} \cup E_{0\,m,n} \right)$ is a kind of "super-potential evaporation", i.e. the potential evaporation that would occur if there was always water available when there is energy available for evaporation AND if there was always energy available when there is water available. We honestly recognize that this concept is not completely easy to understand... and we agree on the need to be able to explain clearly with words what our index does. This is why we propose a new formulation below.

First, let us remind that we initially wished to use exclusively the amount of synchronous P and $E_0$ (in mm/yr) in the regression equation (and in fact, this is what yielded the best results in terms of R²):

$$synchronous\ P - E_0\ amount = \sum_{m=1}^{12} \left( P_{m,n} \cap E_{0\,m,n} \right)$$

Unfortunately, this amount is highly correlated with the annual precipitation sum (and it is not good practice to have correlated independent variables in a regression equation). While this was not a problem for most of the catchments in our dataset, we wanted to avoid it. Moreover; in terms of graphical representation, this positive correlation of $\sum_{m=1}^{12} \left( P_{m,n} \cap E_{0\,m,n} \right)$ with $P_n$ hides the fact that more synchronicity tends to reduce streamflow.

As you noticed, we mentioned in the appendix of the paper some alternatives indices. But we forgot one. If we look for a straightforward interpretation, there are two main alternatives:

$Alt_1 = \frac{\sum_{m=1}^{12}\left( P_{m,n} \cap E_{0\,m,n} \right)}{P_n}$ represents the proportion of annual precipitation which is the most accessible to potential evaporation ( $0 \leq Alt_1 \leq 1$). We mention this alternative in the appendix (Eq.6);

$Alt_2 = \frac{\sum_{m=1}^{12}\left( P_{m,n} \cap E_{0\,m,n} \right)}{E_{0\,n}}$ represents the proportion of annual potential evaporation which is the most accessible to precipitation ( $0 \leq Alt_2 \leq 1$). This is the alternative we forgot to mention.

Returning to the two examples that you mention in your review, the first one (*P = 5 mm every month and PET = 40 mm every month*) would result in $Alt_1$ = 1 and $Alt_2$ = 0.125; which corresponds to your

comment (100% of the precipitation is easily accessible to evaporation). Your second example would result in $Alt_1 = 0.125$ and $Alt_2 = 1$, which corresponds to your comment (most of the precipitation is inaccessible to evaporation).

Using the two indices separately is a possibility, but it makes things less simple and straightforward. Using only $Alt_1$ in the regression is possible, but the problem of the graphical representation remains (the correlation between annual precipitation amounts and $Alt_1$ hides the effect we want to show). Using only $Alt_2$ in the regression is possible, it solves the problem of the graphical representation, but because the annual values of potential evaporation are not very variable, we find ourselves again with two correlated explanatory variables in the regression.

We thought that it would be possible to combine both indices into one:

$$New\ \Lambda = Alt_1 * \bar{P} + Alt_2 * \overline{E_0} = \frac{\sum_{m=1}^{12}\left(P_{m,n} \cap E_{0_{m,n}}\right)}{P_n} * \bar{P} + \frac{\sum_{m=1}^{12}\left(P_{m,n} \cap E_{0_{m,n}}\right)}{E_{0_n}} * \overline{E_0}$$

This 'new lambda' would represent, in mm/yr, the sum of neutralizable precipitation and neutralizable potential evaporation. Any increase of the sum should favor evaporation over streamflow.

We tested this new lambda and found that it is able to explain the annual streamflow anomalies almost as well as the old one and it is visually as satisfying: we plot for example streamflow anomalies and synchronicity anomalies below (using all 4 122 catchments and all 162 005 station-years).

[Figure]

*Figure 1 : graphical comparison of the old and the new synchronicity indices on the entire dataset*

For all these reasons, we propose to modify the $\Lambda$ in the revised version. If this new version is easier to explain and understand, we believe it is worth introducing it.

**2.    Analyzing your suggestion for a synchronicity index**

Thank you very much for your recommendation (λ-New). We did implement it but found that it is in fact linearly correlated with $Alt_1$:

$$\frac{\sum_{m=1}^{12} max\left(P_{m,n}-E_{0_{m,n}},0\right)}{P_n} = 1 - \frac{\sum_{m=1}^{12}\left(P_{m,n} \cap E_{0_{m,n}}\right)}{P_n}$$

The demonstration is as follows:

$$P_n = \sum_{m/P_{m,n}>E_{0_{m,n}}} P_{m,n} + \sum_{m/P_{m,n}\leq E_{0_{m,n}}} P_{m,n}$$

$$= \sum_{m\,/\,P_{m,n}>E_{0_{m,n}}}\left(P_{m,n}-E_{0_{m,n}}+E_{0_{m,n}}\right) + \sum_{m\,/\,P_{m,n}\leq E_{0_{m,n}}} P_{m,n}$$

$$= \sum_{m\,/\,P_{m,n}>E_{0_{m,n}}}\left(P_{m,n}-E_{0_{m,n}}\right) + \sum_{m\,/\,P_{m,n}>E_{0_{m,n}}} E_{0_{m,n}} + \sum_{m\,/\,P_{m,n}\leq E_{0_{m,n}}} P_{m,n}$$

Since

$$\sum_{m=1}^{12}\left(P_{m,n} \cap E_{0_{m,n}}\right) = \sum_{m\,/\,P_{m,n}>E_{0_{m,n}}} E_{0_{m,n}} + \sum_{m\,/\,P_{m,n}\leq E_{0_{m,n}}} P_{m,n}$$

And

$$\sum_{m\,/\,P_{m,n}>E_{0_{m,n}}}\left(P_{m,n}-E_{0_{m,n}}\right) = \sum_{m=1}^{12} max\left(P_{m,n}-E_{0_{m,n}},0\right)$$

We come to the above-mentioned conclusion.

---

## Author Response (AR1)

This document summarizes the main changes introduced in the second version of the paper. We have taken into account all remarks, and modified in-depth both the text, the figures and the synchronicity index to address as much as possible the constructive critics of the reviewers. Note that we do not reproduce here the answers that were published online (https://doi.org/10.5194/egusphere-2025-414-AC1 and https://doi.org/10.5194/egusphere-2025-414-AC2), we do however reiterate our thanks for the time taken by the reviewers to analyze our work.

**Reviewer 1**

The overview overlooks many large-sample hydrological studies that have already pointed (and quantified) clear links between precipitation seasonality (relative to PET or T seasons) and (mean) annual streamflow rates:

Jawitz, J. W., Klammler, H., & Reaver, N. G. F. (2022). Climatic asynchrony and hydrologic inefficiency explain the global pattern of water availability. Geophysical Research Letters, 49, e2022GL101214. https://doi.org/10.1029/2022GL101214

Padrón, R. S., Gudmundsson, L., Greve, P., & Seneviratne, S. I. (2017). Large-scale controls of the surface water balance over land: Insights from a systematic review and meta-analysis. Water Resources Research, 53(11), 9659-9678.

Berghuijs, W. R., Sivapalan, M., Woods, R. A., & Savenije, H. H. (2014). Patterns of similarity of seasonal water balances: A window into streamflow variability over a range of time scales. Water Resources Research, 50(7), 5638-5661.

**References were added**

In addition, it may be useful to point out that Potter et al. (2005) concluded something that opposes the main findings of the current manuscript. Namely, that rainfall seasonality was not reflected in the mean annual water balance

We added the following sentence in section 4.1:

"In the case of Australia, where streamflow anomalies are clearly negatively correlated to the synchronicity index anomaly ( $\Lambda$ ), it is interesting to mention the surprising conclusion of Potter et al. (2005) who wrote that "the inclusion of seasonally varying forcing alone was not sufficient to explain variability in the mean annual water balance", and it is likely that this conclusion was an artefact of the index chosen to describe synchronicity. »

Also note that more asynchronicity indices exist, for example, in papers listed above, but also other works such Willmott, C. J., & Feddema, J. J. (1992). A more rational climatic moisture index. The Professional Geographer, 44(1), 84–88.

I am sorry, but I did not find any synchronicity related info in the modified moisture index of Willmott and Feddema.

The methods should more clearly explain the synchronicity function. The  $\cap$  and  $\cup$  operatators are not extremely widely used in hydrology, and could use a clearer explanation. In general, a visualization, such as provided in the 2018 study that is referenced tom would be helpful. Technically the work also does not explicitly test for phase shifts (only quantifies some indirect effects of that, so the work should reconsider its title.

We replaced the  $\cap$  and  $\cup$  operators by "min" and "max"., and we added a figure.

We modified the title, it now reads "Time shift between precipitation and evaporation has more impact on annual streamflow variability than evaporation"

The choice for a particular example is completely arbitrary and not explained. It is fine to provide a "random" example but then provide to explain the method, and not halfway the results.

The example was moved up to complement our explanation of the method.

The writing of the paper could benefit from fewer sections and less bullet points, and writing it as a more fluent story (I guess this is also partly personal taste, but I believe this may benefit readers of HESS(D) so please at least consider it).

We removed all bullet points.

Figures & subscripts

We did try to save space, but the figures remain spread over several pages.

Why remove all the catchments with reservoirs? Would testing this also across more human-impacted reservoirs not make the study more relevant (either by showing the findings apply to a wider range of conditions or by showing the contrasts in behaviors)?

We would need a large database of human-impacted catchments which would also provide information on the way these reservoirs are managed, and we do not have it. We believe it makes sense to first analyze the pseudo-natural hydrology before discussing how management modifies the natural signal.

The study talks about elasticities (which are defined as % change in response per % in driver), but uses sensitivities (which express mm/y change in response per mm/y in driver). The paper reflects on this use of language but why is does it choose to have this inconsistency between the terms and definitions, and not use conventions?

We agree with you on the difference to be made between elasticities and sensitivities in the "classic" literature. But on the other side, it has become quite common to "extend" the use of the word "elasticity", and we believe that having it in the title allows a clearer identification of the topic of the paper.

L41: The wording of this sentence suggests this is the only way of assessing climate impacts, which is not the case. For example add the word "can", and the issue would be solved.

We modified the whole sentence, which now reads:

"To assess the impact of climate change on water resources, hydrologists aim to quantify the amount of change in catchment flow when climatic conditions vary. The ratio between changes in streamflow and climate is formally defined as the climate elasticity of streamflow (Schaake and Liu, 1989). »

L146: define if your aridity index is PET/P or P/PET.

done

Figure 1: Fix subscript labels

done

Figure 4: add hypen following R^2. Do something to make the overlap in markers more clear.

We added quantile lines to show where the most density lies

Figure 5: x markers are hard to read

Markers were modified

Figure 6: Fix labels (p-valuye, E\_0). The Figure also has an inefficiently large size for something that could be displayed much smaller (of the font was adapted

Labels were fixed.

**• Reviewer 2**

**Issues related to the choice of the synchronicity metric**

We do not want to repeat here the answer we made online, but we want to explain a slight change that was introduced since, in order to simplify further the description of the index.

We mentioned in the online answer that, for us, the simplest way to characterize the "easily accessible water amount" during a year was:

$$S1(n) = \sum_{m=1}^{12} min(P_{m,n}, E_{0_{m,n}})$$
 Eq. 1

But S1 cannot be introduced in the regression formula because  $\Delta S1$  and  $\Delta P$  are too correlated

Two alternatives exist, but they are non-dimensional, comprised between  $0\ \mathrm{and}\ 1$ :

$$\lambda 2(n) = \frac{\sum_{m=1}^{12} \min(P_{m,n}, E_{0_{m,n}})}{P_n}$$
 Eq. 2

$$\lambda 3(n) = \frac{\sum_{m=1}^{12} \min(P_{m,n}, E_{0_{m,n}})}{E_{0_n}}$$
 Eq. 3

The problem we found with  $\lambda 2$  was that it yielded a constant value (equal to  $\bar{P}$ ) for many arid catchments, where for most of the years  $\frac{\sum_{m=1}^{12} \left(P_{m,n} \cap E_{0_{m,n}}\right)}{P_n} = 1$  because  $P_{m,n} \ll E_{0_{m,n}}$ .

Also, we do not wish to use a non-dimensional quantity in the regression, we wish to introduce a quantity in mm/y and this is why we need to re-dimensionalize it:

$$S2(n) = \lambda 2(n) * \bar{P} = \frac{\sum_{m=1}^{12} min(P_{m,n}, E_{0_{m,n}})}{P_n} * \bar{P}$$

$$S3(n) = \lambda 3(n) * \overline{E_0} = \frac{\sum_{m=1}^{12} min(P_{m,n}, E_{0_{m,n}})}{E_{0_n}} * \overline{E_0}$$
 Eq. 5

But S3 behaves similarly as S1 (clearly because the  $\frac{\overline{E_0}}{E_{0n}}$  ratio is always close to 1), and the issue of having highly correlated values of  $\Delta S3$  and  $\Delta P$  reappears.

This is why we propose to use the geometric average of  $\lambda 2$  and  $\lambda 3$

$$\lambda 4(n) = \sqrt{\lambda 2(n) * \lambda 3(n)}$$
 Eq. 6

Then renormalizing can be done in several ways, we used average precipitation  $\overline{P}$ :

$$S4(n) = \lambda 4(n) * \bar{P}$$
 Eq. 7

Let's now consider an example, with a catchment that has average annual P of 240 mm/y (20 mm/month) and average annual  $E_0$  of 240 mm/y (20 mm/month).

Year 1 is very dry:  $P_n = 5$  mm/month and  $E_n = 20$  mm/month

$$\Lambda_1 = \frac{\sum_{m=1}^{12} \left( P_{m,n} \cap E_{0_{m,n}} \right)}{\sqrt{P_n * E_{0_n}}} * \bar{P} = \frac{5}{\sqrt{5 * 20}} * \bar{P} = \sqrt{\frac{5}{20}} = 0.5 * \bar{P} = 0.5 * 20 * 12 = 120 \text{ mm}$$

Year 2 is very wet: P = 35 mm/month and E = 20 mm/month

$$\Lambda_2 = \frac{\sum_{m=1}^{12} \left( P_{m,n} \cap E_{0_{m,n}} \right)}{\sqrt{P_n * E_{0_n}}} * \bar{P} = \frac{20}{\sqrt{35 * 20}} * \bar{P} = \sqrt{\frac{20}{35}} * \bar{P} = 0.76 * \bar{P} = 0.76 * 20 * 12$$

$$= 181.4 \text{ mm}$$

If we consider that the catchment alternates between  $\Lambda_1$  and  $\Lambda_2$ , then

$$\bar{\Lambda} = 150.7$$

$$\Delta \Lambda_1 = -30.7 \ mm$$

$$\Delta \Lambda_2 = 30.7 \ mm$$

Line 32: Hydrological year definition – I am pleased to see that a single hydrological year definition has not been used for the entire world. However, one definition for the Northern Hemisphere and another for the Southern Hemisphere leaves plenty of scope for key periods of synchronicity between P and PET within a year to encounter the arbitrary start or end of the hydrological year. Why not used a hydrological year defined at each catchment based on the month with the lowest monthly Q as the start of the hydrological year? If the authors wish to keep the current two definitions, then please add to the Supplementary Material results of a comparison of the overall results with the two hydrological years versus individual catchment hydrological years. Alternatively, an explanation of why the results aren't expected to change due to hydrological year definition would be appropriate to add in the Notations section.

We looked in the original datasets and we found that most if not all CAMELS dataset authors were recommending the above cited hydrological years

Moreover, a hydrological year based on the month with lowest Q is not adapted for catchments with large snow contribution, where the month with lowest Q will often fall right in the middle of the snowpack accumulation.

Line 103: "seasonality of rainfall" – I think you mean "synchronicity between precipitation and potential evaporation" here.

The sentence was updated. It now reads:

« In this paper, we aim to improve the prediction of streamflow elasticity by introducing anomalies in synchronicity between precipitation and potential evaporation as a predictor, alongside variability in rainfall and potential evapotranspiration. Our study is based solely on data analysis using linear regression models. »

Line 140: I agree that re-computing the PET with a common equation is a good idea. It would be good to add to this paragraph some examples of the PET equations used in the datasets to give a sense of the diversity of PET that they contain.

We added the following sentence:

« In the original datasets, potential evaporation was computed with a variety of different formulas (Makkink, Morton, FAO-56, Penman-Monteith, Hargreaves, Oudin, etc.). »

Figure 1 Caption: You mentioned that the catchments beyond the orange line are leaky and beyond the blue line are gaining. Another possible interpretation is that there are errors in the P, Q and PET data that become apparent in this Turc-Budyko plot. I think it would be good to acknowledge that data errors could be causing some of the unexpected points in these two plots.

We naturally agree with you, but as modelers, we usually try not to criticize the data, because we know too many modelers who keep accusing the data for the shortcomings of their own models. We added the following sentence:

« (note that in a few cases, data uncertainties might also cause catchments to be beyond the limits) »

Equation 4: The first term on the right-hand side should be eQ/P, not eQ/Eo.

**Corrected**

Lines 199 - 200: correct the synchronicity variable name. It should not be V as used in this sentence in four places.

We modified the sentence as follows:

"Where  $\Delta Q_n$  (respectively  $\Delta P_n$ ,  $\Delta E_{0n}$ ,  $\Delta \Lambda_n$ ) represents the deviation from the mean annual value (anomaly) for variable Q (respectively P,  $E_0$ ,  $\Lambda$ ) in mm/y.  $e_{Q/P}$ ,  $e_{Q/E_0}$  and  $e_{Q/\Lambda}$  represent the elasticity of streamflow with respect to P,  $E_0$ , and  $\Lambda$  (dimensionless)".

Table 3: One option here would be to use the Adjusted R^2, which adjusts the R^2 value based on the number of parameters used in the equation. This would make the Adjusted R^2 value more comparable across models with a different number of variables included. A commented is made later on that adding an extra variable is expected to increase R^2, which is true. The adjusted R^2 is designed to take that issue into account.

We changed the R2 values and present now the adjusted R2 values (the world average decreases consequently from 0.70 to 0.67 for the three-parameter regression, and from 0.63 to 0.61 for the two-parameter regression).

---

## Author Response (AR2)

| Antony, | 01.09.2025 |
|---------|------------|
|         |            |

Dear Editor,

I did include all the small corrections suggested by reviewer 2 and modified the title.

I also updated the references.

Thank you again for handling our paper.

Vazken Andréassian